# Moral Distress of Intensive Care Nurses: A Phenomenological Qualitative Study Two Years after the First Wave of the COVID-19 Pandemic

**DOI:** 10.3390/ijerph192215057

**Published:** 2022-11-16

**Authors:** Debora Rosa, Loris Bonetti, Giulia Villa, Sara Allieri, Riccardo Baldrighi, Rolando Francesco Elisei, Paola Ripa, Noemi Giannetta, Carla Amigoni, Duilio Fiorenzo Manara

**Affiliations:** 1Istituto Auxologico Italiano-IRCCS, Piazzale Brescia 20, 20149 Milan, Italy; 2Department of Nursing, Nursing Research Centre, Ente Ospedaliero Cantonale (EOC), 6500 Bellinzona, Switzerland; 3Department of Business Economics, Health and Social Care, University of Applied Sciences and Arts of Southern Switzerland, 6928 Manno, Switzerland; 4Center for Nursing Research and Innovation, Vita-Salute San Raffaele University, 20132 Milan, Italy; 5Ospedale San Giuseppe Gruppo Multimedica, Nursing Degree Course, University of Milan, 20122 Milan, Italy; 6School of Nursing, UniCamillus—Saint Camillus International University of Health and Medical Sciences, 00131 Rome, Italy

**Keywords:** COVID-19, ethics, nursing, qualitative research

## Abstract

Background: The COVID-19 pandemic has imposed great pressure on healthcare facilities, exposing healthcare professionals to various challenges that may result in the onset of moral distress, a condition of psychological distress caused by the inability to act as it would be most morally appropriate. The purpose of this research was to investigate the experience lived by nurses who worked in an intensive care unit during the COVID-19 pandemic. Methods: This is a phenomenological study using interpretative phenomenological analysis. Sixteen nurses who worked in the COVID-19 Intensive Care Unit of Northern Italian Hospitals from January to April 2022 were selected through purposive sampling. Data on experiences, thoughts, and symptoms were collected through semi-structured interviews with in-person and remote modalities. Results: Five themes and fourteen subthemes emerged from the study. The themes are: (1) pride, isolation, and fear; (2) teamwork and organisation; (3) moral/ethical aspect; (4) true heroes; and (5) dignity. Conclusions: This study highlights the impact of the COVID-19 pandemic on intensive care unit nurses. It has emerged that the risk of moral distress is high among healthcare workers in the front line of the fight against the virus. This condition should be avoided and managed through early psychological interventions, sharing of experiences, and a good organization that supports decision-making and professional well-being.

## 1. Introduction

The SARS-CoV-2 pandemic has been a major health problem for the past two years. From the first wave to date, the World Health Organization reports 516,922,683 confirmed cases and 6,259,945 deaths [1].

Italy was the first country in the Western world to be hit by the COVID-19 pandemic, with the first case reported in Lombardy. The sudden spread of the pandemic caught health systems around the world unprepared, leading to serious organisational and resource shortcomings—both human and material—in tackling the health emergency [2,3]. Intensive care units (ICUs) were the most stressed wards because the COVID-19 infection was causing severe respiratory failure [4].

In all countries, the lack of up-to-date strategic plans of personal protective equipment (PPE), trained personnel, and a sufficient number of intensive care beds has led to severe distress among healthcare personnel, with the onset of psychological distress and burnout [5,6]. 

A possible source of this discomfort among professionals is ethical, as health workers, in particular, nurses, were no longer able to provide care according to the usual standards due to the exponential number of patients to be treated in a short time [7,8]. This situation made decision-making more difficult for health professionals, who sometimes had to choose whom to treat, as neither material nor human resources were sufficient for everyone. The situation was so serious that some professional associations had to issue guidelines to help professionals make their decisions [9]. Thus, difficulty in making decisions, lack of resources, fear of infecting oneself and one’s loved ones, isolation, changes in end-of-life care, and restrictions on hospital visits increased the occurrence of ethical conflicts among healthcare professionals during the pandemic [7,10,11,12].

Moral distress (MD) is frequently reported in intensive care settings [13,14]. Considering the impact of the COVID-19 pandemic on healthcare systems, this situation has increased the likelihood of developing MD among intensive care nurses. MD can have negative effects on nurses’ well-being and can increase the likelihood of leaving the profession and the intention to leave the ICU [15,16]. In most cases, the impact of MD on the practitioner arouses negative feelings and affects the psychological state of the person, reflecting on the quality of the care provided and resulting in reduced professional fulfilment [17]. During the pandemic, many nurses not experienced in intensive care were transferred to the ICU due to the lack of staff. The situation was probably even more stressful for those nurses due to their inexperience in intensive care.

Simonovich et al. reported that nurses’ MD during the pandemic was related to fear, frustration, helplessness, and guilt [18]. Similar results were reported by Bayerle et al. [19] Silverman et al. [20], in their qualitative study, identified seven MD-related themes in nurses caring for patients with COVID-19: (a) a lack of knowledge and uncertainty about how to treat a new disease; (b) being overwhelmed by the depth and breadth of COVID-19 disease; (c) fear of exposure to the virus leading to suboptimal care; (d) the adoption of a team-based model of nursing that caused intra-professional tensions and miscommunication; (e) policies to reduce viral transmission (visitation and IPR policy) that prevented nurses from assuming their caring role; (f) the practice of crisis care standards; and (g) the management of scarce medical resource. In intensive care, Donkers et al. [21] reported that the main source of MD during the pandemic was “inadequate emotional support for patients and their families”. 

Although some studies have already investigated MD during the pandemic, as we can see, the results vary from one context to another [22,23,24]. Therefore, this phenomenon should be studied more thoroughly. Furthermore, in our context, there are no studies that have investigated MD during the COVID-19 pandemic in the ICU setting.

Exposure to ethical conflicts may have fostered MD among nurses [25]. MD was first defined by Jameton in 1984 as “a condition that occurs when the nurse makes a judgement by virtue of a given situation and is faced with obstacles that prevent the next action to be taken” [26]. In 1993, Jameton further elaborated on the concept of MD, distinguishing it into initial and reactive MD. In initial MD, the nurse experiences frustration, anger, and anxiety in the face of obstacles when obstructed by institutions and in interpersonal conflicts over professional values [27]. Reactive MD, on the other hand, is the direct consequence of the failure to process feelings arising from initial MD. Several researchers have expanded and redefined these ideas by introducing the concepts of MD and moral residue [28,29]. These authors describe the moral residue as the persistent feelings and personal disagreements resulting from the MD that continue even after the event is over. It is this residual discomfort that can cause much damage over time, especially when a person is repeatedly exposed to morally distressing events. Epstein and Hamric [29] called this phenomenon the “crescendo effect.” Even if people recover to some extent from these events, repeated exposure accumulates over time. In fact, these events can affect the moral conscience of professionals, cause a great deal of distress, and put future actions at risk. For this reason, we decided to conduct this study, which aimed to explore the MD experience of Italian ICU nurses during the COVID-19 pandemic.

## 2. Materials and Methods

### 2.1. Study Design

We adopted a phenomenological study design using interpretative phenomenological analysis (IPA) [30]. Phenomenology is a qualitative research approach committed to examining how people make sense of their major life experiences. IPA takes a hermeneutic, idiographic, and phenomenological stance, allowing a nuanced explication of participants’ experiences, and is well suited for concepts that are complex and ambivalent, such as the topic under scrutiny [30]. The study was conducted following the consolidated criteria for reporting qualitative research checklist (COREQ) [31].

### 2.2. Sampling

The research team chose a convenience sample of 16 nurses working in the ICU. Inclusion criteria incorporated knowledge of the Italian language and the nature of the study, having lived through the experience covered by the study, signing of the informed consent, age of 22 years or older, and having worked in ICU during the first wave of the COVID-19 pandemic. Having clinically relevant cognitive dysfunction was an exclusion criterion. We continued recruiting patients for the study until we reached a data saturation point [30,32]. The construct of saturation in qualitative research is complex and questionable; however, the research team thought that the understanding of the information retrieved could give sufficient depth to shed light on the topic under exploration [33].

### 2.3. Data Collection

Data were collected through semi-structured interviews performed remotely due to the situation. The semi-structured interviews consisted of open-ended questions (Table 1. Semi-structured interview flow). The interviews were conducted by a female PhD nurse researcher who is expert in qualitative research and who was not involved in patient care. In addition, the research team consists of three undergraduate students from the Bachelor of Science in Nursing course; two master’s degree nurses in Nursing and Midwifery, one of whom has a degree in psychology; two PhDs in Nursing and Public Health; and an associate professor in Nursing.

The semi-structured interview technique was designed to encourage the participants to share stories or experiences in response to a broad, data-generating question and to primarily facilitate an interaction letting participants tell their own stories in their own words [30]. Open-ended questions were used to encourage participants to reflect on their own experiences and to generate affluent and detailed narrative answers from the participants about their professional experiences during the first wave of COVID-19. 

Each interview was conducted in a quiet location to ensure no interruptions during the interview and was audio recorded, subjected to verbal consent from the interviewee, to facilitate subsequent transcription and data analysis using a numeric identifier to ensure anonymity. Once transcribed, the interview was deleted from the device used for the recording. 

### 2.4. Data Analysis

For the data analysis the researcher did not use any software. Data were analysed in accordance with the principles of the IPA, which involves the identification of meaning units, sub-themes, and themes in adherence to the language of the speakers [30]. Two researchers independently started the analysis, investigating in-depth one case after another. In the case of disagreement, the two researchers returned to the original texts of the interviews and to the field notes and reformulated the shared themes. The first step of the IPA involves immersing oneself in the original data. At this stage, it is important to read and re-read the transcripts, make margin notes, create a summary list of the margin notes, develop emergent themes, and look for connections among them. Once each interview was individually analysed, the process was then extended to scan the entire set of transcripts for a full listing of theme summaries, grouping of theme summaries, recoding transcripts with overall themes, and finalising the list of themes with extracts [30]. 

### 2.5. Study Rigour

The accuracy and rigour of the data were ensured by the cross-checking of two researchers: the first researcher transcribed the collected data, and the second researcher listened to the audio recording while reading the transcription to ensure that it accurately reflected the words of the interviewee. Any information that could make the interviewee recognisable was edited to protect his or her identity. For this purpose, pseudonyms were used instead of names of people, cities, streets, or organisations [34].

The criteria for promoting trustworthiness referred to credibility, transferability, reliability, and confirmability [33,35]. Strategies were adopted to ensure reliability: prolonged researcher engagement in the study, interviews and multiple member-check notes, and triangulation [33,35]. Researcher characteristics were also important. They had to be responsive and adaptive to changing circumstances, showing professional immediacy, sensitivity, and the ability to clarify and summarise [33]. Verbatim quotes were included throughout the analysis to give voice to the participants so that readers could trace the research team’s interpretations, demonstrating sensitivity to the context [36]. 

The results obtained are not generalisable but potentially transferable to similar settings.

### 2.6. Ethical Approval

The work was carried out in accordance with the code of ethics of the World Medical Association (Declaration of Helsinki). Ethical approval was obtained from the Institutional Review Hospital Board and the Hospital Directions. The interviews were conducted in a private and quiet space in the hospital while respecting the patients’ privacy. Confidentiality was assured, and the participants were free to leave the study at any time. The investigators assigned each participant a number to hide their identities. The research team gave every participant an information sheet with detailed information about the study and it was read carefully and explained by the investigators before the interview. The interviewers allowed patients to read the written consent, reflect, and decide before giving their informed consent.

## 3. Results

The interviews were conducted from January to April 2022. The sample consisted of 16 nurses, of whom nine (six women and three men) were interviewed in person and seven (three women and four men) were interviewed remotely. These interviews were video-audio recorded and transcribed verbatim. The interviews lasted between 24 and 63 min. 

Table 2 describes the sample’s socio-demographic characteristics. The researchers identified five main themes: (1) pride, isolation, and fear; (2) teamwork and organisation; (3) moral/ethical aspect; (4) true heroes; and (5) dignity of the affections (Figure 1).

### 3.1. Pride, Isolation, and Fear

The lack of positive social relationships is a factor that can contribute to the development of MD [6]. Furthermore, the absence or decrease in self-care are indicators of MD [37]. Interviewees, through the knowledge they developed in the various departments where they worked, changed their lifestyles. In particular, they isolated themselves to avoid infecting the people who lived with them and fragile family members (the elderly and children). This isolation led to the realisation that there was often no time to take care of themselves. Fortunately, many of the interviewees experienced the difficulties they encountered as an incentive to improve professionally. The researchers identified the following sub-themes: isolation and loneliness; no time to think about myself; fear for loved ones; and difficulty as a stimulus.

#### 3.1.1. Isolation and Loneliness

In this sub-theme, the nurses described the change in their lifestyle as a result of the precautions and knowledge acquired on the ward.


*“Both me and my husband …”–“I tried to keep a certain distance but I do not have a huge house. I was also distancing myself from hugging my little girl, physical contact was not very easy.” (ID.3)*



*“On the one hand, positive things … one’s own space, one’s own autonomy … on the other hand, however, you have a sense of isolation … in the sense that I am used to arriving home and having a chat with someone … finding myself alone … without being able to go out or anything … it was a mixture of emotions … good and bad ….” (ID.5)*


#### 3.1.2. No Time to Think about Myself

A recurring aspect in the interviews is that of the nurses’ difficulty in thinking and caring for themselves during the COVID-19 pandemic, due both to the lack of time and accumulated fatigue and to the emotions they experienced, which led to apathy and abandonment. Some interviewees reported feeling drained and lacking strength at the end of the working day, mainly due to the accumulated tension during the shift. This caused the nurses to experience their work with anxiety and their return home as another opportunity to reflect on what had happened on the ward, thus neglecting their focus on themselves and their own well-being. This also caused a sense of not being able to guarantee the same quality of care, a typical feeling related to MD. 


*“For the first month and a half, the days for us were all the same. This meant that we never stopped and there was no time, for me and for us, to think.” (ID.13)*



*“I would come home and there would be a moment when I would try to relax, as if all the tension of the day would collapse, and there, however, I would often […] cry […]. I … just needed to let off steam when I came back from there [the hospital].” (ID.12)*


#### 3.1.3. Fear for Loved Ones

The fear of infecting significant persons (parents, partners, children, etc.) was identified by all respondents, especially as linked to a lack of knowledge of the disease during the first wave. What was clear was the high contagiousness and risk of developing severe symptoms, especially for elderly people and/or people with comorbidities.


*“At home, I was a bit terrified because it was a bit like the plague, you did not know how to deal with it … Then, obviously, working in a hospital, you had to take some precautions at home as well.” (ID.2)*



*“It had mixed emotions … I found myself suddenly having to leave home to protect my family, as my father had a heart condition and during the first wave of COVID-19, I had left home and found a rented flat.” (ID.5)*


#### 3.1.4. Difficulty as a Stimulus

According to some of the professionals interviewed, the pandemic was also experienced as a formative moment in which the nursing professional was able to prove himself/herself. From both an emotional and learning perspective, it was expressed how much this period also influenced the interviewed nurses in a “positive” way. 

During the COVID-19 pandemic, several of the nurses interviewed had to change their working environment, moving from medium-to-low-intensity wards to the ICU, thus needing time to acclimatise and learn nursing procedures.


*“The situation taught me a lot, in the sense I think I wouldn’t have arrived today with all the knowledge I have now if it hadn’t been for COVID-19. I quit a lot of safety during COVID-19.” (ID.5)*



*“It is not trivial to have the courage to face something you do not know.” (ID.11)*


As in other studies, one of the aspects that caused MD the most was the inadequacy of care at the end of life, due to the high number of deaths per day and the absence of relatives, which often caused a lonely death. Although nurses tried to make up for this lack with video calls, this certainly created great frustration and a sense of inadequacy, increasing the experiences of MD.

### 3.2. Teamwork and Organisation

During the first wave of COVID-19, some protective factors developed that may have helped contain MD [6]. Respondents claimed that teamwork, emotional support, and work support allowed professionals to cope with the situation more easily. Sharing a common difficult moment seems to have united the healthcare staff on the ward even more.

During the interviews, the nurses emphasised the importance of teamwork, thanks to which, in particular, they were able to experience confrontation, which was a further stylus for professional growth.

The following sub-themes emerged in this theme: mutual support in difficulty and impact against a wall.

#### 3.2.1. Mutual Support in Difficulty

The COVID-19 ICU experience was described as positive, engaging, challenging, constructive, and characterised by the presence of a common goal. The health professionals involved developed intra- and inter-mutual support.


*“We were all there fighting the same enemy. Before, we were a little more competitive, in conflict, for example compared to other colleagues from other departments, instead now we were all there, we supported each other.” (ID.1)*



*“With the doctors and anaesthesiologists there was a good group, a good collaboration, on the other hand we were in the same boat.” (ID.2)*


The professional noted during the first wave that an extreme effort was required of the ward nurses who had to provide care, causing them discomfort. The nursing coordinator was described by most of the nurses interviewed as an active part of professional development and an understanding point of referral. 


*“The problem I found was in managing the constantly changing shifts, the wards were changing and having to keep asking for extreme availability …, at a certain point you realise that availability was taken for granted”–“If I could have … managed more effectively ….”–“The nurses were committed, they accepted the requests and you realise that every time you ask for something you are a bit on eggshells, you have to be more careful because for me there was a lot of insecurity left from this staff point of view … keep changing the colleague.” (ID.4)*



*“We knew, however, that we could ask her [head nurse]. This I will never stop saying. Even from a psychological point of view … she gave us a great hand.” (ID.12)*



*A nurse interviewed highlighted the care taken by the head nurse, who tried hard to alleviate stress and also took care of the psychological and meditative components.*



*“The head nurse was very attentive, especially in the beginning. She would help us, she would push us, she would even send us online yoga classes (sneers) … she would make up anything and everything.” (ID.10)*


#### 3.2.2. Impact against a Wall

The professionals felt inadequate, as they had to improvise courses of treatment in relation to the absence of validated knowledge about the disease. 


*“I remember that we were given the instruction booklet on how to operate the fans. We felt inadequate … let’s just say we improvised a bit from that point of view.” (ID.1)*



*“Worry, at first worry. First because you didn’t know what it was or how to deal with it and you saw so much confusion around. And the fact of seeing so much, so much confusion around automatically spilled over onto us … At a certain point there were protocols to follow and you found yourself having to be helpless, not being able to do anything at all and seeing a life passing away there, in front of you, without being able to do anything, because nothing worked … so that really leaves a mark on you.” (ID.3)*



*“It was quite an impact against a wall … in the sense that we found ourselves doing the jobs that we were not ready to do….” (ID.7)*


The majority of those interviewed claimed to have felt abandoned, especially in the early stages of the pandemic. Indeed, the shortage of materials, the need to change wards due to the absence of sick colleagues and working with different and unfamiliar professionals did not make the nurses feel secure in their care. Often, the only fixed point was the ward head nurse who tried to create and maintain a climate of solidarity and collaboration within the ward. The nurses highlighted the perceived lack of support from the hospital, which allegedly further accentuated feelings of loneliness and professional discomfort. In addition, incidents of excessive shifting and constant ward changes were described. Some nurses dwelt on the lack of organisation of the healthcare system in dealing with emergencies, highlighting, in particular, the deficiencies in terms of staff and equipment.


*“I absolutely did not share this position, then, show yourself; at a certain point you put on your gown for once, dress up, but come and show yourself … this made us feel abandoned….” (ID.1)*



*“I understand that there were no resources … but it could have been better managed, in my opinion….” (ID.7)*


The main limitations identified by the interviewees related to the lack of knowledge about the ICU, the new COVID-19 disease, and professional inexperience. An element highlighted by several interviewees was the presence within the team of professionals from different operating units, which made it difficult to standardise knowledge, skills, and working methods. The feeling of working with unskilled colleagues was a typical aspect that caused MD. 


*“Having never worked in such a context, with such a tough and tough care complexity, I found myself having to learn in very few days everything there was to know in order to be able to manage a patient with high care complexity.” (ID.13)*



*“In my opinion, the biggest difficulty was that we came from different realities, without knowing each other, by the way, so standardising our skills was complicated and I don’t even know if we succeeded in the end.” (ID.10)*


### 3.3. Moral/Ethical Aspect

MD occurs when the nurse is aware of the morally appropriate action to be taken, but cannot implement it due to institutional obstacles (lack of time and/or support), medical power, legal limitations, inadequate knowledge [27]. The interviewees highlighted the presence of ethical dilemmas concerning the quality of care and treatment provided and the accompaniment of the patient at the end of life. In fact, for the professionals, respect for the person’s dignity at all stages of the illness was of primary importance, but they sometimes found themselves having to compromise with the doctors.

The following sub-themes were identified in this theme: learning how to compromise; protecting the suffering of the patient; and end of life, dignity of the person, and ethical/moral suffering.

#### 3.3.1. Learning How to Compromise

The interviewees, in the following sub-theme, brought out their thoughts in relation to the care that was implemented, which they did not always agree on and which consequently generated discomfort in the nurses who worked in the relevant operational unit. Another factor that emerged was the lack of knowledge of the clinical condition, which affected the type of care to be implemented, causing the professional to compromise in order to try to ensure the health of the patient as much as possible. 


*“I had a fit of tears, because decisions had been taken which were completely at odds with what we were experiencing, we were pointing out that we were in difficulty with certain aspects….” (ID.1)*



*“To compromise … to think about certain things”–“obviously terminal patient, non-invasive ventilation, face all bruised … on morphine to try to keep her calm; when asked: ‘excuse me, but hasn’t anyone tried to ask a doctor following her if something concrete can be done?’ It doesn’t change the destiny, rather we put a less invasive aid, but we try not to let her die with a wounded face’ …” (ID.5)*


#### 3.3.2. Protecting the Suffering of the Patient

This sub-topic identified the suffering of the nursing professional related to the nursing process, particularly in the accompaniment at the end of life.


*“How hard it was to see the dead person put in the sack with that soaked sheet, to close the sack; to know that no one would ever see him again, no family or anyone. And can you imagine the short pause if it had happened to you! To me!” (ID.3)*


#### 3.3.3. End of Life, Dignity of the Person, and Ethical/Moral Suffering

The nurses interviewed perceived the absence of protection of the dignity and suffering of the assisted person, who was subjected to invasive therapeutic procedures, even when there was no need.


*“Because what has always been valid for me, regardless of the COVID-19, is the respect of dignity … both when one is alive but even more so when one is in a terminal phase”–“to try to ensure the life they had as best as possible …” (ID.5)*


### 3.4. True Heroes

Some protective factors of MD have been identified in this theme. In fact, in the opinion of the interviewees, the real heroes of this pandemic were the patients who struggled with all their strength, facing such a dramatic moment as the disease, in a condition of loneliness and uncertainty due to the unfamiliarity with the disease caused by the new COVID-19. In their interviews, the nurses repeatedly emphasised the commitment and strength they put into helping patients overcome the illness and feel less alone. The sub-themes are clinging to life with strength; you stay in my heart; and relying on health personnel.

#### 3.4.1. Clinging to Life with Strength

The nurses expressed admiration for the patients admitted to the COVID-19 ICU. The patient’s strength to cope with the disease was a great testimony and lesson for the interviewees.


*“I was amazed by the patients. The first day I set foot in that tent I thought “not a single person here gets out, nobody makes it” and instead they really showed strength in sticking to life. I feel so much admiration for them, for how they dealt with this dramatic moment in their lives.” (ID.8)*



*“Patients have taught us so much.” (ID.10)*


#### 3.4.2. You Remain in My Heart

Even small improvements spurred the nurses on; this positivity was returned to the patients, becoming a source of hope for them.


*“To the patient [who] tells you ‘I don’t know what happened, I mean, I don’t understand, but I see that I’m slowly getting better’, we tried to give him hope, saying ‘you were really sick before, now you’re slowly getting over it from the critical phase and I’m happy for you, because my efforts have finally paid off’.” (ID.11)*


#### 3.4.3. Relying on Health Personnel

Patients with COVID-19 and their relatives blindly relied on the care and assistance provided by ICU nurses. Patients and their relatives had expectations of the healthcare staff regarding their own and their loved one’s state of health, entrusting their hopes so that the trust placed in them would not be disappointed.


*“Relatives trusted us, nurses and doctors, without hesitation.” (ID.10)*


### 3.5. Dignity

The suffering and discomfort that can result from MD is manifested through fear, frustration, guilt and helplessness [38]. The relatives of COVID-19 patients had to cope with bereavement differently than before the emergency, without having time for a final farewell and without the possibility of performing all the social functions related to the loss of a loved one, making the grieving process more difficult. During the pandemic, communication between relatives and patients took place exclusively at a distance, making the communication process difficult, especially in cases of poor compliance or an inability to communicate.

During the interviews, some nurses emphasised the dignity of the relatives and the fear they experienced due to the patient’s clinical condition. The sub-themes identified were video calling as a window to the world and the dignity of relatives.

#### 3.5.1. Video Calling as a Window to the World

Video calling was a continuous communication tool during the stay of COVID-19 patients, allowing relatives to receive information regarding the clinical course of their loved ones. During the interviews, the nurses emphasised the importance and emotional value of video calls, a fundamental tool to allow the sharing of moments, both joyful and dramatic, between relatives and patients. At times, nurses played a central role, acting as an intermediary between the two parties, especially in the presence of communication difficulties.


*“The relatives were far away [but] we managed to use video calls, either to communicate with relatives or to have patients communicate directly with relatives. I must say that with every video call I attended, I almost cried (sneers). It was emotional. [Attending the video calls].” (ID.10)*



*“Strange, emotional to communicate [with relatives] instead of patients because they were intubated. They couldn’t speak, they had maybe a voice wire and on the phone, they couldn’t be heard, it was you who actually had to then communicate with the relatives. You were immersing yourself in the family, in the intimate [and] in the difficulties that [every] family [had] to face.” (ID.10)*


#### 3.5.2. Dignity of Relatives

The dignity of relatives was mentioned several times by the nurses interviewed. Relatives were forced to cope with their loved one’s illness at a distance, without being able to provide support and reassurance, having to rely totally on the health personnel. Some of the nurses interviewed witnessed their renunciations and sacrifices. 


*“The relatives have taught me a tremendous lesson. Seeing the enthusiasm of the first video call was pure joy. I used to say to myself ‘how can they be satisfied with that?’ The relatives had dignity, the ability to give up seeing their loved one, to accept depersonalised telephone information.” (ID.9)*


## 4. Discussion

This study aimed to explore the MD experience of Italian ICU nurses during the COVID-19 pandemic. The interviews were conducted sometime after the experience of the pandemic, and therefore the experiences reported are already the result of an elaboration of the operators involved.

For the interviewees, the main causes of MD were a lack of knowledge about the ICU and the new COVID-19 disease. This led the respondents to experience fear, anxiety [39,40], and fear of not feeling up to the demands [39]. Other causes of MD were lack of experience and not being used to dealing with critical clinical situations on a daily basis [41,42]. Furthermore, several interviewees pointed out that dealing with a high number of deaths on the same day was a further obstacle, as they rarely witnessed patients dying in their home departments.

The fear of infecting themselves and their loved ones led the nurses to isolate themselves in order to protect their relatives [40,42], exacerbating their discomfort [42]. Respondents also failed to mention much more serious and disabling symptoms such as anxiety, fear, depression, compassion fatigue, burnout, and Post Traumatic Stress Disorder, which have been described in the literature [5,39,40,43]. 

In the interviews conducted, no effective strategies for the prevention and treatment of MD emerged, although many have been identified in the literature, such as relaxation techniques, psychotherapy, sharing experiences and emotions with colleagues, group counselling, defining a strategic health emergency management plan, education and training of practitioners, organisational management, defining universal guidelines, exercise, maintaining social relationships, and seeking hobbies and support from friends and family [5,39,40,41,42].

The literature highlights other forms of strategies defined as non-functional, such as alcohol abuse and eating a poor and unbalanced diet [44], absenteeism, and high turnover [43]. Such strategies did not emerge from the data analysis; on the contrary, in some cases, respondents preferred to go to work, skipping a few days off rather than staying at home to think about colleagues in distress. In fact, nurses also expressed difficulty in completely relaxing and taking their minds off the emergency situation once at home [5,44]. 

One of the aspects frequently mentioned by the nurses interviewed was the hospital organisation in response to the medical emergency. The main organisational obstacles relate to the management of scarce resources, admission criteria to the ICU, and unclear and, in some cases, contradictory priorities and guidelines [7,41]. The presence of a team comprising professionals from different operational units led to difficulties in sharing knowledge and working methods in a unified manner. 

Regarding interventions for the management of MD, according to Miljeteig et al. and Shen et al., certain early interventions are prioritised [41,42]. These may be training, the presence of clinical prioritisation guidelines, and the allocation of nurses according to their competencies. Such interventions described as effective in the literature were not reported by the interviewees, who instead reported a lack of such interventions.

The presence of good leadership also seems to guarantee support and the development of resilience [39]. In fact, the role of the nursing coordinator was fundamental for some interviewees because he or she became an active part of the growth process and a point of reference for the group. The nursing coordinator made it possible to alleviate the state of stress by improving the psychological components [22].

Collaboration between professionals within the operating units was perceived as a positive experience during the COVID-19 pandemic [45]. Despite the difficulties encountered, the nurses dwelled on some positive aspects, first and foremost the teamwork, expressing gratitude to their colleagues for the climate created, for the mutual growth, and for the help received during difficult moments. The collaboration between the nurses allowed the integration of skills, knowledge, and abilities, resulting in better results in patient care. The nurses, due to the lack of certainty regarding the pandemic, experienced very strong moods, and some of them emphasised how much teamwork contributed to maintaining a state of well-being. Falcó-Pegueroles et al. [7] showed that measures such as maintaining good communication with the team reduced the exposure to conflict between nurses and doctors, adding that the lack of division of tasks and roles made work difficult.

In initial MD, the nurse experiences frustration, anger, and anxiety when faced with situations when he/she sees himself/herself hindered by institutions and in interpersonal conflicts over professional values. Reactive MD is a direct consequence of the non-processing of feelings born in the initial MD [46]. As a form of stress, this affects nurses, and although Seyle [47] states that stress can be a motivating influence for change, MD reflects a negative response in relation to problems in the working environment. 

An important organisational choice, implemented in an attempt to reduce the spread of infection, was to prevent caregivers from entering hospital wards. This entailed the need to find alternative strategies that would still allow family members to come into contact with hospitalised patients in order to ensure effective communication that is fundamental to the well-being and satisfaction of patients and families [48]. The nurses interviewed attributed video calls to a central role in ensuring that the relationship between patients and their relatives was maintained, although they represented a greater emotional burden for nurses, who often felt extremely involved. 

In the results of this study, it is evident that the nurses interviewed found it necessary to talk about patients and family members as relevant figures within their own experiences during the pandemic. There were not many articles in the literature in which nurses reflected on the experiences of patients and family members. In our study, the participants wanted to emphasise the strength with which patients clung to life, the recognition that patients and relatives showed towards professionals, and the dignity of relatives who trusted professionals and completely entrusted the lives of their loved ones to them.

MD is a cross-cutting phenomenon that affects all health professionals [6,49]. In a qualitative study, it was highlighted that after one year, physicians who worked in ICUs during the pandemic spontaneously adopted different responses to manage MD: avoidance, acquiescence, resistance, and reinterpretation. Avoidance of morally distressing situations emerged as an initial and provisional self-protective response. Acquiescence and resistance are responses that generate an internal struggle to silencing personal values and accepting the situation. Reinterpretation was the only response that allowed doctors to restore a sense of moral integrity, recognising their limited degree of freedom in situations and finding new possible ways to be good doctors in circumstances [50]. Although many studies have been published on this subject, few have been carried out after two years.

### Limitations

One of the limitations of this paper was that we were not able to conduct lengthy interviews with the participants due to the strong emotions they displayed during the interviews. The interviews may have been influenced by the participants’ need not to recount in depth the disturbing memories of COVID-19 and the time elapsed since the experience. A second limitation may have been the fact that the interviews were conducted almost two years after the first wave. This may have influenced the participants’ recollection. However, during the interviews the participants were able to distinguish very well between their experiences during the first and second wave and the others that occurred, for example, after the start of the vaccination campaign, where the epidemiological situation and consequently the impact on intensive care had changed considerably. 

Another possible limitation was the difficulty in taking notes and data related to the nonverbal communication of the interviewee, such as motor, psychobiological and verbalization aspects with field notes during videoconference interviews. Finally, another limitation was that for some nurses the experience of working in intensive care was limited to the period of the first wave.

## 5. Conclusions

During the analysis of the interviews, it emerged that most of the nurses experienced the first wave of the COVID-19 pandemic with an emotional and psychological malaise that is still present two years later. In addition, the lack of adequate training in the management of the patients greatly affected the quality of care provided, and the situation was made worse by the lack of material, lack of knowledge of the clinical condition resulting from COVID-19, lack of support from the organisations, excessive workload, and disagreement with the therapeutic process implemented and inadequate end-of-life support. All of these factors generated discomfort in nursing professionals, who saw their ethical and moral values disappear, especially with regard to the protection of the patients’ suffering and the protection of their dignity. In addition to what has already been mentioned, the professionals were afraid of infecting their loved ones, considering that their work in the COVID-19 ICU put them in direct contact with people suffering from the disease. This factor resulted in a self-imposed isolation of the nurses from their loved ones, generating discomfort in the professionals. This may have increased the MD, as nurses perceived that they were not able to guarantee the same quality of care as before, due to the lack of both material and human resources, due to the high number of patients, the lack of knowledge, and the feeling of hopelessness that this caused. Consequently, we can state that the nurses who participated in the study experienced MD associated with the first wave of the COVID-19 pandemic. The analysis showed that in an emergency situation with inadequate preparation, the nurse should be protected first and foremost so that the delivery of care is adequate and the health of the professional is guaranteed.

As in other studies, one of the aspects that caused MD the most was the inadequacy of care at the end of life, due to the high number of deaths per day and the absence of relatives, which often caused a lonely death. Although nurses tried to make up for this lack with video calls, this certainly created great frustration and a sense of inadequacy, increasing the experiences of MD [51].

Intervening through increased moral education, seminars for nurses, and discussion of ethical/moral conflicts with colleagues and other health professionals can help foster collegial relationships and inter-professional collaboration [52,53]. Furthermore, institutional hospital leadership may consider encouraging the involvement of nurses in developing policies and guidelines for ethical decision-making in the application of unnecessary therapeutic interventions [54]. The implementation of diaries in which the practitioner can report episodes that have generated distress may also improve both the physical and mental health of the nurse [55].

A hypothesis for future study may be the investigation of how COVID-19 has encouraged nurses to leave the profession in relation to MD and an inadequate ethical climate and what actions can be taken to ensure that this does not happen in a similar situation. 

## Figures and Tables

**Figure 1 ijerph-19-15057-f001:**
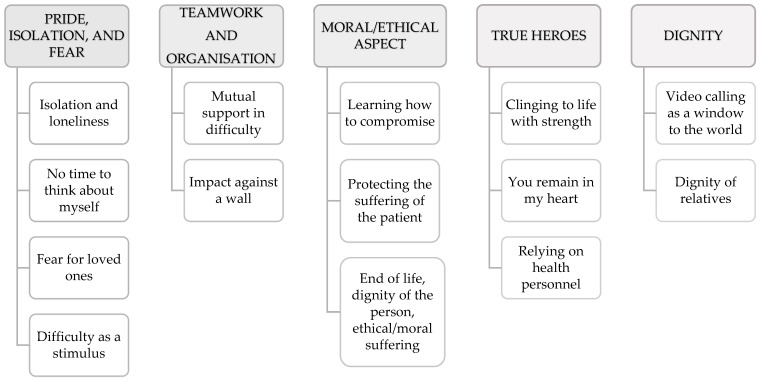
Themes and subthemes identified.

**Table 1 ijerph-19-15057-t001:** Semi-structured interview flow.

Questions
1. Could he talk about you?
2. How would you describe yourself as a nurse?
3. What was the period of the COVID-19 pandemic like for you?
a. What was your experience in your private life?
b. What was your experience in your working life?
4. Could you tell us what kind of experience was the COVID-19 pandemic for you with respect to your moral or ethical perspective?
5. What influenced your decisions from an ethical or moral perspective during the period of the COVID-19 pandemic?
6. Could you tell us about situations in which you disagreed with your team about what was ethically or morally correct during the period of the COVID-19 pandemic?
7. How did this experience influence you?

**Table 2 ijerph-19-15057-t002:** Participant characteristics (N = 16).

Female	9
Male	7
With partner	7
Single	9
Age (Mean)	38.03 (Range 25–54)
Years working experiences	14.52 (Range 1–33)
Years working experience ICU (Mean)	1.77 (Range 3 months to 8 years)

## Data Availability

Not applicable.

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
