# Peer review of "Moral Distress of Intensive Care Nurses: A Phenomenological Qualitative Study Two Years after the First Wave of the COVID-19 Pandemic"

_ijerph, 2022, doi:10.3390/ijerph192215057_

Round 1

Reviewer 1 Report

I thank the authors for the opportunity to review this interesting article. However, take into account the following recommendations and answer the questions raised:

• In the data collection, they state that some interviews were conducted remotely and others in person. You must detail how many were done in person and how many by videoconference. How did you conduct the videoconference interviews? Did you encounter limitations during its realization? It is important to add a table with the questions asked in the interview and the areas of interest to investigate.

• In data analysis it is important to describe whether they used computer software.

• I understand that the authors intend to study the experience of the nurse who worked from the first wave of Covid until the beginning of 2022. It is complex to analyze a phenomenon of these characteristics retrospectively since experience has surely changed with the pandemic. . Why did they wait almost two years to analyze the experience of nurses in the first wave? Going back to the anguish experienced two years ago is not easy and experiences can be mixed.

• In table 1, the marital status of the participants is not detailed. I am surprised that Years working experience ICU (Mean), is 1.77. Why did you not include more experienced nurses? Your experience during the first wave of the pandemic is probably directly related to your work experience, please justify your choice.

• The font size of figure 1 is very small and when enlarging the document it does not look good, please increase the size.

• I consider that the results should group the categories within the same section of the topic. In this way, the section is given more consistency, that is, remove the category subtitles and write it in a single section. I do not think it contributes too much to differentiate so many sections. I recommend that participant narratives be written in italics.

• The section presented as a conclusion seems more like a continuation of the discussion. You should not contradict your results with those of other studies.

Thank you very much. All the best

Author Response

Dear reviewer

we made our best in order to improve the quality of the paper. 

  • In the data collection, they state that some interviews were conducted remotely and others in person. You must detail how many were done in person and how many by videoconference. How did you conduct the videoconference interviews? Did you encounter limitations during its realization? It is important to add a table with the questions asked in the interview and the areas of interest to investigate.

Answer: Thank you for the suggestion. We added the information in the introduction and limitations

  • In data analysis it is important to describe whether they used computer software.

Answer: Thank you for the suggestion. We added the information.

  • I understand that the authors intend to study the experience of the nurse who worked from the first wave of Covid until the beginning of 2022. It is complex to analyze a phenomenon of these characteristics retrospectively since experience has surely changed with the pandemic. . Why did they wait almost two years to analyze the experience of nurses in the first wave? Going back to the anguish experienced two years ago is not easy and experiences can be mixed.

Answer: Thank you for your suggestion. We agree with you that this aspect can had been a limit of our study. Therefore, we added a statement in the limitation setting about it. However, we would like to underline that the participants have a clear picture of their experience during the first and second wave and they were able to distinguish their experience in the first and second wave, compared to subsequent ones.

  • In table 1, the marital status of the participants is not detailed. I am surprised that Years working experience ICU (Mean), is 1.77. Why did you not include more experienced nurses? Your experience during the first wave of the pandemic is probably directly related to your work experience, please justify your choice.

Answer: Thank you for the suggestion. We added the information

  • The font size of figure 1 is very small and when enlarging the document it does not look good, please increase the size.

Answer: Thank you. We have corrected the figure 1 as you suggested

  • I consider that the results should group the categories within the same section of the topic. In this way, the section is given more consistency, that is, remove the category subtitles and write it in a single section. I do not think it contributes too much to differentiate so many sections. I recommend that participant narratives be written in italics.

Answer: Thank you, we revised the results, in particular 1 theme and 2 sub-themes. We modified the quotations in italics as you suggested

  • The section presented as a conclusion seems more like a continuation of the discussion. You should not contradict your results with those of other studies.

Answer: Thank you, we revised the discussion

Reviewer 2 Report

Overall: The background and introduction to this article was very well-written. It set up the stage nicely for a study on moral distress, which was the aim of the study, line 107. However, the results reflected everything but moral distress. Very interesting information, but not related to the primary study aim. Then in the discussion, the aim changed, line 445. Moral distress is the inability to do the ethically correct course of action, however, what was described throughout the results were not connected to moral distress in a clear way. More elaboration on how these findings are connected to moral distress would be helpful. Unless the results just aren't there, then we can't conjecture. See Latimer et al., 2022 in Palliative and Supportive Care for an example of a qualitative approach conducted in social workers about MD during pandemic.

Introduction: this is not the first study to discuss moral distress in ICU during the COVID pandemic.

Methods: The length of interviews were very short, discussing why may be helpful for readers to understand the limitation. Details about the interviewees were provided in lines 131-132, however, this was never tied into why it mattered when it came to interpreting the results, which I would expect given the reflexivity involved in IPA. The table of demographics needs aligned left and smaller. The table of themes were a nice illustration to sum results, but some text was cut off (e.g., pride, isolation, and...). 

Lines 103-105 need to be removed- it's unclear of its intent and has negative implications on nursing care/ nurses.

Conclusions: Are nicely written, but do not reflect the results as they connect to moral distress.

Author Response

Dear reviewer we made our best in order to improve the quality of the study

Overall: The background and introduction to this article was very well-written. It set up the stage nicely for a study on moral distress, which was the aim of the study, line 107. However, the results reflected everything but moral distress. Very interesting information, but not related to the primary study aim. Then in the discussion, the aim changed, line 445. Moral distress is the inability to do the ethically correct course of action, however, what was described throughout the results were not connected to moral distress in a clear way. More elaboration on how these findings are connected to moral distress would be helpful. Unless the results just aren't there, then we can't conjecture. See Latimer et al., 2022 in Palliative and Supportive Care for an example of a qualitative approach conducted in social workers about MD during pandemic.

Answer: Thank you for the suggestion. We added the information in the results

Introduction: this is not the first study to discuss moral distress in ICU during the COVID pandemic.

Answer: thank you for the suggestion. We revised the introduction

Methods: The length of interviews were very short, discussing why may be helpful for readers to understand the limitation.

Answer: Thank you. We added this as a limit. The range was 23 to 64 minutes, probably some interviewees did not feel like exploring certain themes in depth because they were considered emotionally charged. We underlined this as a limitation in the first paragraph.

Details about the interviewees were provided in lines 131-132, however, this was never tied into why it mattered when it came to interpreting the results, which I would expect given the reflexivity involved in IPA.

Answer: The data collection section (line 130-135) reports the characteristics of the research team as suggested by the COREQ guidelines used for this study. In addition, the data analysis section reports the methods of analysis by detailing the methods followed by the researchers.

The table of demographics needs aligned left and smaller. The table of themes were a nice illustration to sum results, but some text was cut off (e.g., pride, isolation, and...). 

Answer: Thank you. We have corrected the table 1 as you suggested

Lines 103-105 need to be removed- it's unclear of its intent and has negative implications on nursing care/ nurses.

Answer: Thank you for the suggestion. We removed the lines

Conclusions: Are nicely written, but do not reflect the results as they connect to moral distress.

Answer: Thank you for your suggestion. We added some more sentences that connect our results to MD. We hope to have correctly understood your request.

Round 2

Reviewer 1 Report

After introducing the suggested changes, I consider that the article can be published